# Geo-Agriculture: Reviewing Opportunities through Which the Geosphere Can Help Address Emerging Crop Production Challenges

**Chris Pratt** [1,*]**, Kate Kingston** [2]**, Bronwyn Laycock** [3] 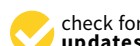**, Ian Levett** [3] **and Steven Pratt** [3]

[1] School of Environment and Science, Australian Rivers Institute, Griffith University, Kessels Road, Nathan, QLD 4111, Australia

[2] School of Environment and Science, Environmental Futures Research Institute, Griffith University, Kessels Road, Nathan, QLD 4111, Australia; kate.kingston@griffith.edu.au

[3] School of Chemical Engineering, University of Queensland, St Lucia Campus, Sir Fred Schonell Drive, QLD 4072, Australia; b.laycock@uq.edu.au (B.L.); i.levett@uq.edu.au (I.L.); s.pratt@uq.edu.au (S.P.)

\* Correspondence: c.pratt@griffith.edu.au; Tel.: +61-7-3735-3605

**Abstract:** The agricultural sector faces looming challenges including dwindling fertiliser reserves, environmental impacts of conventional soil inputs, and increasingly difficult growing conditions wrought by climate change. Naturally-occurring rocks and minerals may help address these challenges. In this case, we explore opportunities through which the geosphere could support viable agricultural systems, primarily via a literature review supplemented by data analysis and preliminary-scale experimentation. Our objective is to focus on opportunities specifically relating to emerging agricultural challenges. Our findings reveal that a spectrum of common geological materials can assist across four key agricultural challenges: 1. Providing environmentally-sustainable fertiliser deposits especially for the two key elements in food production, nitrogen (via use of slow release N-rich clays), and phosphorus (via recovery of the biomineral struvite) as well as through development of formulations to tap into mineral nutrient reserves underlying croplands. 2. Reducing contamination from farms—using clays, zeolites, and hydroxides to intercept, and potentially recycle nutrients discharged from paddocks. 3. Embedding drought resilience into agricultural landscapes by increasing soil moisture retention (using high surface area minerals including zeolite and smectite), boosting plant availability of drought protective elements (using basalts, smectites, and zeolites), and decreasing soil surface temperature (using reflective smectites, zeolites, and pumices), and 4. mitigating emissions of all three major greenhouse gases—carbon dioxide (using fast-weathering basalts), methane (using iron oxides), and nitrous oxide (using nitrogen-sorbing clays). Drawbacks of increased geological inputs into agricultural systems include an increased mining footprint, potential increased loads of suspended sediments in high-rainfall catchments, changes to geo-ecological balances, and possible harmful health effects to practitioners extracting and land-applying the geological materials. Our review highlights potential for 'geo-agriculture' approaches to not only help meet several key emerging challenges that threaten sustainable food and fiber production, but also to contribute to achieving some of the United Nations Sustainable Development Goals—'Zero Hunger,' 'Life on Land,' and 'Climate Action.'

**Keywords:** agriculture; climate change; contamination; drought; fertiliser; geosphere

---

## 1. Introduction

Inorganic rocks and minerals dominate the foundations of agricultural production. They comprise, on average, >95% of the soil's dry mass [1] and have been demonstrated to be critical to several key soil processes including water retention [2,3] and nutrient cycling [4–9].

Several researchers have advocated 'geo-agricultural' approaches to help support cropping practices. In 2002, van Straaten published 'Rocks for Crops: Agrominerals of Sub-Saharan Africa.' The author detailed the chemical composition of the most commonly encountered bedrock types in 48 African countries with potential identified for using these sources for nutrient supplementation as well as moisture retention in agronomic enterprises in that region [10]. Following on from van Straaten's (2002) foundational work, Edwards and Lim [11] reviewed the potential for land application of crushed volcanic rocks to mitigate against climate change. The weathering of these particular rock types results in the formation of stable mineral carbonate phases, effectively 'mopping up' $CO_2$ from the atmosphere. Edwards et al. [11] advocate this approach particularly in the tropics, where weathering rates are rapid. They note that the delivery of macronutrients to plants is potentially associated with an agricultural benefit in these settings. Importantly, these authors recognised possible downsides for using crushed volcanic rocks in tropical agriculture, including the associated mining footprint and increased sedimentation in catchments during heavy rainfall. Beerling et al. [12] also reviewed the potential for land application of crushed volcanic rocks to sequester atmospheric $CO_2$ and recognised additional inherent benefits in the approach including a supply of key nutrients (Ca and Mg) to plants. However, these authors noted that methods for quantifying the climate change offset of this strategy are urgently needed.

These reviews present only glimpses of the full potential of geological amendments to agricultural land. Inclusive of the topics discussed above, we identify four key areas where the geosphere could significantly rejuvenate agricultural production systems: (1) offering new sustainable fertiliser reserves of key nutrients, (2) mitigating nutrient contamination in farming catchments, (3) protecting agri-ecosystems during environmental stresses—particularly drought, and 4) mitigating climate change impacts caused by agricultural activities.

In this paper, we review the current state of knowledge relating to each of these themes and highlight opportunities for new directions and further development, particularly relating to emerging agricultural challenges. This is the first review, as far as we are aware, that comprehensively collates the multi-faceted contributions through which the geosphere could help rejuvenate agricultural practices at the wider level. Our main objective is to demonstrate that many of the emerging challenges to agriculture might be addressed by considering ways that the geosphere—specifically applications of rocks and minerals to agricultural land—can be integrated with established farm management practices. Importantly, a number of the themes that we address in this paper highlight opportunities to realise benefits beyond the agricultural sector alone. Efforts to minimize pollution, foster resilient landscapes, and combat emissions contributing to climate change are all wide-reaching societal aspirations. We identify that 'geo-agricultural' approaches could directly help support three of the seventeen Sustainable Development Goals prioritized by the United Nations, such as 'Zero Hunger,' 'Life on Land,' and 'Climate Action'.

## 2. Methods

### 2.1. Literature Review

The review was conducted following a systematic approach using Google Scholar and Web of Science databases. Initial key word searches included combinations of the terms: 'agriculture', 'climate change', 'contamination', 'drought', 'fertiliser', 'geology', 'minerals', and 'rocks'. Based on the authors' research and professional experience, four key 'geo-agriculture' opportunity themes were developed and searched,

which include: 1. New fertiliser prospects (key search terms included combinations of: 'crustal abundance elements', 'high nitrogen and phosphorus content rocks and minerals', and 'waste nutrient streams'). 2. Environmental contamination prevention (key search terms included combinations of: 'zeolites', 'clays', 'bentonite', 'oxides/oxyhydroxides', 'allophane', 'hydrotalcite', 'ammonia/ammonium', 'nitrate', 'nitrite', 'phosphate', and 'eutrophication'). 3. Drought resilience (key search terms included combinations of: 'water holding capacity', 'zeolites', 'clays', 'bentonite', 'specific surface area', 'silicon', 'potassium', 'light reflectance', and 'temperature'), and 4. Climate change mitigation (key search terms included combinations of: 'enhanced weathering', 'silicates', 'olivine', 'iron oxides', 'zeolites', 'clays', 'carbon dioxide', 'methane', 'nitrous oxide', 'ammonia', and 'agricultural climate change mitigation').

### 2.2. Modelling Moisture Retention Capacity for Zeolite-Amended Soils

Although this paper is predominantly a review, it contains data analysis and experimental work to support some of the concepts discussed. We use data from a report by Kingston [13] to model soil moisture retention rates in two agricultural soils (Vertisol and Planosol—World Reference Base for Soil Resources) amended with zeolites. The glasshouse experiment tested the water retention capacity of the soils in 1 L pots sown with tomato plants conducted over four months. Zeolite amendments were applied in a geometric sequence of eight levels ranging from 5 to 200 t/ha. Moisture retained in the two soils at the completion of the experiment was converted to aerial rates using literature bulk density values for the two soil types under field conditions [14,15] assuming an active zeolite incorporation depth through the topsoil layer (i.e., top 0.2 m). Linear regression models were determined for water retention versus zeolite application for both soil types using SPSS Statistics (Version 25).

### 2.3. Preliminary Reflectance and Surface Temperature Experiment

In a supporting experiment, 20 g of four common agricultural soils (classified as Vertisol, Planosol, Gleysol, and Ferralsol, according to the World Reference Base for Soil Resources) and four prospective geo-amendment materials (zeolite, smectite, vermiculite, and pumice) were oven dried overnight at 105 °C. These materials were placed onto a white paper surface to a depth of 1 cm. Colour space model properties (i.e., light (L), chroma (C), and hue (H)) of the materials were measured using a digital colour sensor (NIX), following internal calibration using the NIX Digital App. The samples were then placed in ambient sunlight for a 2-h period. The surface temperature of all materials was measured by an infrared thermometer (Stanley), which was calibrated to three temperature levels set on a convection drying oven (40 °C, 60 °C, and 80 °C). Temperature measurements were recorded every 30 min and used to calculate an average surface temperature over the monitoring period. Ambient air temperature was recorded using a mercury thermometer. Linear regression analysis was performed on an average surface temperature and colour space properties, using SPSS Statistics (Version 25).

### 2.4. Semi-Quantitative Analysis of Beneficial Mineral Attributes

Toward the end of this review, we present a semi-quantitative analysis of the four key mineral attributes reviewed. Indices were assigned according to the following criteria. 1. Fertiliser potential—the maximum value (outer contour band of web plot) was attributed if the mineral contributes at least one of the six macronutrient elements, median value (middle contour band of web plot) if mineral contributes micronutrient element or pH adjustment. 2. Drought resilience—the maximum value was assigned for minerals that increase moisture-holding capacity, median value scored if the mineral offers available forms of drought-protective elements (Si and K) or temperature protection via heat reflectance. 3. Eutrophication mitigation—given the maximum value if the mineral can sequester N and P compounds, otherwise naught was assigned. 4. Climate change mitigation—the maximum score was assigned for

minerals with the potential to mitigate $CO_2$ and median value assigned for minerals with potential to mitigate lower-magnitude greenhouse gases (GHGs) (i.e., $CH_4$ and $N_2O$), and 5. abundance—the following ranking values were assigned for the main mineral groups: 1. silicates, 2. oxides, 3. carbonates, 4. sulfates, 5. halides, 6. Phosphates, and 7. nitrates. The silicate subset groups were ranked as: 1. neso and ino-silicates, 2. smectite and vermiculite, 3. tobelite, and 4. zeolite. These rankings were then normalized to give the five (i.e., 20%) contour bands in the web plots.

## 3. New Opportunities for Harnessing Geosphere Fertiliser Reserves

### 3.1. Geological Fertiliser Prospects: Overview

The main nutrients plants require are nitrogen, phosphorus, potassium, magnesium, calcium, and sulphur. In lesser amounts, iron, manganese, copper, boron, molybdenum, zinc, sodium, and chloride are essential for crop production. Figure 1 presents typical nutrient element concentrations for a range of geological sources, which are categorised as 'minerals' and 'rocks.'

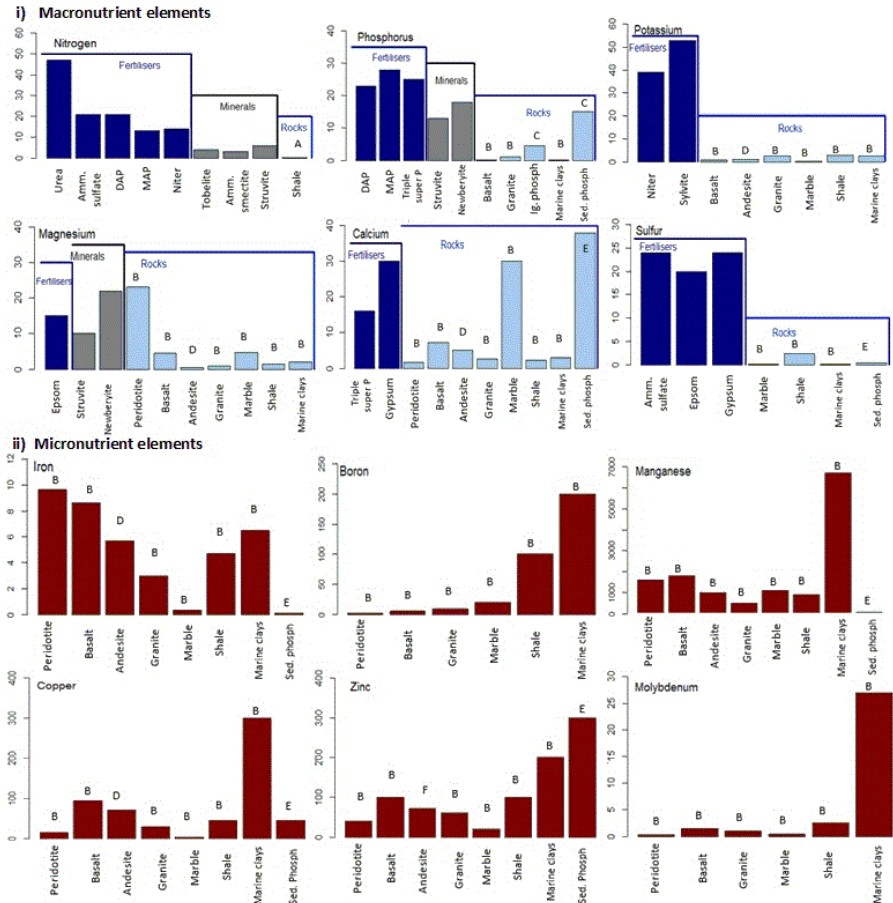

**Figure 1.** Typical concentrations of essential plant (**i**) macro (values in wt %) and (**ii**) micro elements (values in ppm, except iron which is wt %) across a range of prospective geological sources (minerals and rocks) for use in agriculture, with conventional fertiliser contents ("fertilisers") for macro-elements shown for comparison. Fertiliser and mineral concentrations derived from chemical stoichiometry. DAP = diammonium phosphate, MAP = monoammonium phosphate, Amm. Smect. = ammonium smectite, Triple Super P = Triple Super Phosphate, Ig. Phosph. = Igneous Phosphorites, Sed. Phosph = Sedimentary Phosphorites. Reference sources: A = [16], B = [17], C = [18], D = [19], E = [20], F = [21].

For the macro-element nutrients, the geological materials are benchmarked against conventional fertiliser products (Figure 1i). For cases where the conventional fertiliser form is a mineral (e.g., gypsum, sylvite/("potash")), these are aligned into the fertiliser category. Sodium and chlorine sources are not included as these are typically present in surplus concentrations in broad-scale agriculture.

All key nutrient elements are enriched in several of the listed geological source materials (Figure 1). Geological materials are the principal sources for most plant nutrient elements, particularly micronutrients, which are derived from concentrated base metal reserves—typically hydrothermal deposits—and then industrially-processed to separate the target elements.

Most of the key plant nutrient elements outlined in Figure 1 are not in short supply within recoverable reserves at the Earth's surface where locally abundant, direct application of concentrated geological materials to agricultural land might offer a viable alternative for using industrially-processed formulations in efforts to decrease processing and transportation footprints. However, we are at a crossroads in the way we harness the two most important crop plant elements. These include nitrogen (N) and phosphorus (P). We now examine approaches through which the geosphere could rejuvenate our current agricultural management of these elements.

## 3.2. Geological N Prospects

Conventional N fertiliser sources, while having been critical in delivering increased crop yields to feed growing populations, are associated with some serious negative consequences. Both chemical and organic N fertilisers (e.g., manures, compost) are significant sources of N losses to the environment via volatilisation of ammonia [22], which is a nuisance odour and indirect greenhouse gas (GHG), emission of the direct GHG and ozone-depleting nitrous oxide [23], and nitrate leaching [24]. Conventional chemical N fertilisers also exert a substantial $CO_2$ footprint from energy inputs invested in the Haber-Bosch process [25,26]. Moreover, the hydrogen used in ammonium, ammonia, and urea manufacture is derived from fossil methane reserves and it is estimated that about 1.5% of global energy stocks are invested into chemical N fertiliser production [27]. Additionally, in the case of urea, the carbon is fossil in origin. Once land is applied, this carbon—which comprises approximately 20% of the fertiliser mass—is almost entirely labile and constitutes a further environmental impact [28]. Reserves of naturally-occurring N-rich rocks and minerals might offer viable fertiliser supplements to help address some of these issues.

The geosphere accounts for huge proportions of the Earth's N budget—crustal rocks contain almost 99% of fixed N: $10^{21}$ g compared with $10^{19}$ g in the biosphere [29]. Moreover, geological N has been shown to be biologically active with Holloway and Dahlgren [30] reporting that bedrock-sourced N saturated the biological N requirements in forest ecosystems in the US. However, although geological N may be biologically very active, it is unlikely to entirely support plant growth in agronomic contexts where nutrient demands are much higher at about several hundred kg per ha per annum. But geological N could supplement conventional fertilisers.

Holloway and Dahlgren [30] classify N-rich rocks (>10,000 mg/kg) into three groups: 1. high-N silicates, 2. organo-sedimentary rocks, and 3. nitrate mineral deposits. Rocks rich in organic-N associated with high levels of labile carbon are likely not ideal for harnessing as fertiliser sources because of the risk of associated fossil $CO_2$ emission. These are included in the 'Shale' category in Figure 1. Concentrated nitrate deposits—'niter' in Figure 1 have historically been used for global agricultural N supply [31], but their potential for nitrate leaching hampers justification to use these resources in lieu of manufacturing N fertiliser sources by chemically fixing atmospheric N. Rocks containing N-bearing silicate minerals are potentially the most attractive geological N source for complementing established fertiliser products. These typically low-grade metamorphic deposits contain high phyllosilicates (clays, micas) where ammonium was substituted for potassium in the mineral structure. The resulting mineral, tobelite, is the ammonium

end-member of muscovite [32]. Extensive ammonium-enriched tobelite deposits have been recorded in many locations including Japan, Europe, China, and the North Sea [32–34]. Tobelite offers a potentially viable N fertiliser reserve given its N concentrations are reasonably enriched (Figure 1i).

Tobelite's exchange chemistry also offers the prospect of slow release into soil environments, potentially matching plant uptake needs. Synchronising N supply to plant requirements could yield environmental benefits in addition to reducing input costs by preventing rapid mineralisation transformations, which exacerbate N losses to the environment via volatilisation, gaseous emission, and leaching [22,35–37]. While tobelite reserves alone are unlikely to meet widespread N fertiliser needs, other clay minerals with similar $NH_4^+$-exchange capacity are much more ubiquitous. These include the high cation exchange smectites and vermiculite. Extracted deposits of these minerals could offer additional tailored N supply to crops. 'Doping' these minerals with N ions from waste streams (to produce ammonium smectites, Figure 1i) might offer a sustainable approach to future agricultural fertilisation practices. We follow up on this in the 'Environmental Contamination Prevention' section later in this MS.

### 3.3. Geological P Prospects

Phosphorus presents a different challenge to N. Conventional P sources are almost entirely geological but are becoming rapidly depleted [38]. Phosphate rock fertiliser deposits, termed phosphorites, can reach 17% P by weight [39]. Although uncertain, it is estimated that conventional phosphorite reserves, which are concentrated in a handful of countries, may become exhausted within the next 100 years [40]. Consequently, alternative P sources are sought.

Human waste and livestock manures are postulated to be prospective alternatives to rock phosphate [41,42]. However, large-scale manure application to soils can lead to environmental problems such as N losses, trace metal, and organic contamination issues [43,44]. Moreover, manure and waste supply are usually spatially sporadic, which poses logistical challenges. Alternative P-rich geological sources could complement P fertilisation strategies in the coming decades.

Very few naturally-occurring rocks and minerals can match phosphorite for P content. Otherwise, they would already likely be earmarked as viable fertiliser reserves. Yet geological P concentrations are relatively high (>1000 ppm) within the most commonly encountered crustal rocks (i.e., basalt and granite). Some alkali granites have exceptionally high P levels (1% by weight, Figure 1i) and these are commonly distributed along continental mountain zones [45]. Similarly, certain basalts can have P concentrations of 1%, particularly ultrabasic varieties such as nephelinites and tholeiites [46]. Whole rock phosphorus concentrations of >1 wt % are approaching the lower resource quality threshold and these ubiquitous igneous sources might represent viable future opportunities for geological P recovery.

Extracting P from its source rock is an important consideration. Most commercial applications involve the recovery of P using acids or thermal treatment. Given that the mineral form of P is the same across different geological sources (i.e., apatite), there should be no pronounced differences in approaches to extracting P from varying rock types. In some cases, direct soil application of crushed rock phosphorus has been trialed with varied results. Smyth and Sanchez [47] reported soils with high clay and free iron oxides enhanced rock phosphate dissolution and plant P availability, even though Bolland and Gilkes [48] found generally poor uptake of P from rock phosphates in various crop species in Western Australia. Bolland and Gilkes [48] concluded that P release from unprocessed rock phosphates is too slow under most pHs and temperatures for effective uptake by crops.

Rock P availability has been shown to be enhanced via fungal, bacterial, and plant-mediated processes. Azcon et al. [49] found that rock phosphate availability was improved in the presence of mycorrhiza and phosphate solubilising bacteria. Omar [50] reported clear improvements in dry matter wheat yields and P content when fertilised with rock phosphate in the presence of three

fungal strains—*Aspergillus niger*, *Penicillium citrinum*, and *Glomus constrictum*. Xiao et al. [51] found *Aspergillus japonicus* and *Penicillium simplicissimum* to be other mycorrhiza strains capable of solubilising rock phosphate, even though plant availability was not measured. These studies highlight the opportunity for in situ extraction from agricultural regions overlying P-rich bedrock where granules of underlying rock fragments are mixed with the upper productive soil horizons. In these cases, nutrient delivery could be achieved through inoculation with active bacterial and fungal strains [52,53].

Perhaps the most appealing geological P alternative to conventional P reserves is the mineral struvite ($MgNH_4PO_4 \cdot 6H_2O$). Containing 12% P by weight and with its associated constituents of N and Mg, struvite certainly presents as a competitive product when benchmarked against conventional P fertilisers (Figure 1i). There are two main types of struvite deposits: 1. geological—ancient reserves in the Earth's crust and 2. biogenic—produced by concentrating waste streams often via chemical manipulation.

Geological struvite deposits are not widespread but have been detected in localities in the United States, Central America, and South America [54–56]. Another naturally-occurring phosphate mineral, newberyite ($MgHPO_4 \cdot 3H_2O$), which is the main mineral group associated with recently formed guano deposits, has also been detected in older geological sequences [55]. The P content of newberyite (18%) is even higher than that of struvite and, therefore, deposits of this mineral represent a very attractive potential supplement to conventional P sources. Resource quality reserves would need to be mapped and quantified to establish the potential viability and scale of these P mineral reserves. Nonetheless, even if these deposits are viable, they will undoubtedly have limited lifespans similar to conventional rock phosphate reserves.

Struvite precipitated from wastes, by contrast, represents an enticing renewable P resource. Most domestic and agro-industrial wastewater streams carry phosphorus, and it has been estimated that 15–20% of world demand for phosphate rock could be satisfied by recovering P from human waste [57]. However, wastewaters are inherently dilute with total phosphorus concentrations typically below 15 mg/L. Enhanced biological phosphorus removal can be applied to concentrate phosphorus—by at least an order of magnitude—into a sludge stream [58]. Digestion of the sludge then mobilises the phosphorus, and potentially initiates struvite mineralisation, which can even occur spontaneously within treatment infrastructure that creates a nuisance in treatment processes. The opportunity for controlled struvite recovery and use as a fertiliser has been long-recognised [59–61]. However, producing struvite in sufficient quantities for resource extraction continues to be logistically challenging. The main limiting step appears to be economically-viable sources of Mg inputs needed for the mineral structure and to maintain sufficiently high pH (>8.5) for crystallisation. Stolzenburg et al. [62] used Mg oxide by-products from the magnesite industry to feed a struvite crystallisation reactor treating effluent. Aguado et al. [63] used seawater as the Mg source to precipitate struvite from synthetic urine but found that this substrate produced a high volume of CaP products along with the struvite crystals. An alternative, potentially viable substrate for Mg addition to struvite crystallisation is crushed basalt. Basalt is the most commonly-occurring rock at the Earth's surface and commonly contains >5% Mg by weight (Figure 1). It is also latently alkaline, which favours struvite crystallisation.

### 3.4. Geological Fertiliser Prospects: Best Contenders

In summary, the most viable and sustainable geological prospects to support agricultural fertiliser practices appear to be: 1. N-rich clays, whose N content is either naturally-occurring or added to the reactive clay surfaces through contact with N-rich waste streams, 2. unlocking the in situ nutrient content rocks of agricultural soils via inoculation of polymer-encapsulated microorganisms, and 3. recovery of the biomineral struvite, which can be precipitated from a variety of waste streams using other geological materials such as basalt to stimulate precipitation.

## 4. Environmental Contamination Prevention

Fertilisers are designed to release N and P in their most plant-available forms—i.e., $NH_4^+$, $NO_2$, $NO_3^-$, and $PO_4^{3-}$. On the flipside, these compounds, once mobile, are susceptible to runoff and leaching, which results in eutrophication. Agriculture has been identified as a principal contributor to the eutrophication of internationally-significant ecosystems including the Great Barrier Reef [64] and the Great Lakes of North America [65].

The use of geological materials in technologies to mitigate both N and P loss in agricultural settings is receiving increasing attention. Zeolites and clays are ideally suited for binding with ammonium ($NH_4^+$) due to their permanent negative charge [66,67]. Various positively-charged geomaterials including nano-scale aluminium oxides [68], iron oxides/hydroxides [69,70], layered hydroxides [71], and modified clays [72] have been trialed to remove nitrates ($NO_3^-$), nitrites ($NO_2^-$), and phosphates ($PO_4^{3-}$) from agricultural effluents. Allophane, which is a volcanic mineral common to New Zealand and South America, has also been shown to effectively retain P from a variety of waste streams. Although strictly a clay mineral with a net negative surface charge, allophane has been shown to effectively bind with P through ligand exchange complexes [73]. Table 1 summarises literature studies reporting N and P removal from various agricultural streams using geomaterials.

**Table 1.** Performance of various geological minerals removing ammonium, nitrate, and phosphate from waste streams.

| Mineral | Nutrient Targeted | Stream Type | Solution Concentration | Maximum Sorption Capacity | Removal % | Reference |
|---|---|---|---|---|---|---|
| Zeolite (clinoptilolite) | Ammonium | Rice paddy runoff | 14 mg $NH_4^+$/L | 2 g $NH_4^+$-N/kg | 30–50 | [66] |
| Zeolite | Ammonium | Piggery and dairy effluent | 129 mg $NH_4^+$/L | 2 to 9 g $NH_4^+$-N/kg | - | [74] |
| Chitosan-coated smectite | Ammonium | Synthetic effluent | 33 mg $NH_4^+$/L | 4 g $NH_4^+$-N/kg | - | [75] |
| Vermiculite and Smectite | Ammonium | Piggery, poultry, and beef cattle effluent | - | - | 80 | [23] |
| Nano Al oxides | Nitrate | Synthetic effluent | 20 mg $NO_3^-$-N OR total N (unclear)/L | 4 g $NO_3^-$-N/kg | - | [68] |
| Hydrotalcite (hydroxide) | Nitrate | Eutrophied stream | >25 mg $NO_3^-$-N/L | 27 g $NO_3^-$-N/kg | >90 | [76] |
| Hydrotalcite (hydroxide) | Phosphate | Eutrophied stream | >25 mg $PO_4^{3-}$-P/L | 9.8 g $PO_4^{3-}$-P/kg | >90 | [76] |
| Amorphous Fe and Al oxides | Phosphate | Agricultural runoff | 1 to 10 mg $PO_4^3$-P/L | 5 g $PO_4^{3-}$-P/kg | - | [70] |
| Artificial Fe and Al oxides | Phosphate | Eutrophied stream | 120 mg $PO_4^{3-}$-P/L | 4.5 g $PO_4^{3-}$-P/kg | 35 | [77] |
| Allophane | Phosphate | Dairy farm runoff | 10 mg $PO_4^{3-}$-P/L | 3 g $PO_4^{3-}$-P/kg | - | [78] |

Clearly, a range of candidate geological materials can effectively sequester nutrients from paddock runoff (Table 1). However, while technically feasible, one of the main obstacles to engineered approaches to removing nutrients from agricultural runoff is managing the filters once their capacity is exhausted. One novel approach to potentially overcome this barrier is to recycle the exhausted filter media as a soil amendment. All of the geological materials shown to be effective nutrient adsorbents are natural soil constituents in varying amounts. Moreover, these materials exhibit properties beneficial for soil conditioning including high surface areas, which can aid in increasing soil moisture retention and soil microbial activity. Figure 2 shows the broad-scale potential of this approach, which focuses on a nitrogen return to irrigated croplands.

Albeit preliminary, the mass balances in Figure 2 highlight the potential for nutrient cycling using geo-amendments. A 10% nutrient return to paddocks is high enough to be economically attractive to farmers with regard to fertiliser offset costs (which are typically $1500/ha/year). Further potential benefits include a slow-release of nutrients to crops based on the exchange mechanism of nutrient retention by filter media candidates [79]. This feature, combined with the high moisture-retention capacity of the

high surface area filter materials, potentially means less chemical fertiliser requirements with ongoing application of the spent filter media as a soil conditioner.

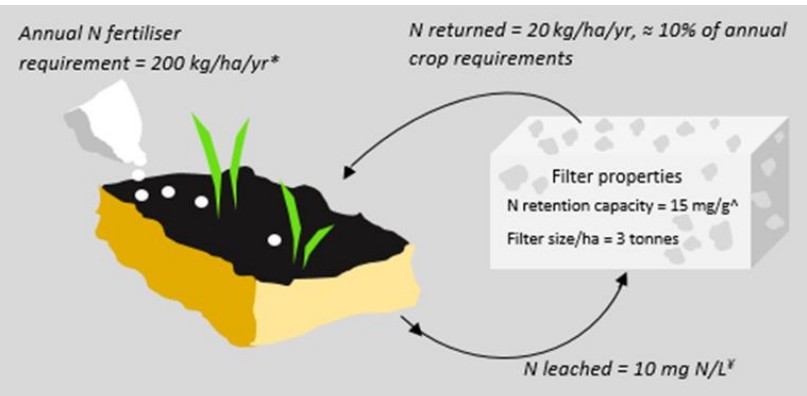

**Figure 2.** Schematic depicting how geo-amendments (e.g., clays, zeolites, or iron hydroxides) could be used to intercept nitrogen (as either $NH_4^+$ or $NO_3^-$) from agricultural runoff and then used as soil amendments for N supply once their filtering capacity becomes exhausted. References for calculation estimates: * from Rowlings et al. [80], ^ = capacity based on average N sorption capacity for zeolites from Wang et al. [66] and hydroxides from Terry [76]. Filter size estimated from performance under a hydraulic retention time of 10 h, based on average irrigation (not rainfall) runoff flow of 3000 $m^3$/ha/year for cane and banana farms in Queensland from Faithful et al. [81] and, assuming bulk density of 1.5 t/$m^3$ for typical packed silicate minerals, the ¥ average of estimate range from Faithful et al. [81] and Wang et al. [66] for agricultural runoff.

## 5. Embedding Drought Resilience into Agricultural Soils

Agricultural landscapes will likely experience increasingly hostile conditions under changing climate [82,83]. One trend of particular concern is a predicted increase in the severity and duration of droughts in dryland cropping regions [84,85].

Manufactured materials, which are most commonly polymers, have been evaluated for their capacity to raise soil moisture retention. These include beads, polyacrylamide and polysaccharide-based hydrogels, and industrial mineral frameworks [86,87]. Although generally effective, manufactured soil amendments entail production costs. Moreover, in some cases, they pose detrimental environmental impacts at the field application stage, especially non-degradable polymers that could be precursors to microplastic pollution [88] and metal frameworks, which can contain high levels of mobile toxic trace metals [89].

Boosting soil organic matter has been adopted as another approach to improve soil moisture retention properties [90]. However, many arid agricultural landscapes struggle to retain healthy soil organic matter stocks [91] and ongoing replenishment of organic materials into the soil may not be practical. Soil inputs of geological materials could help arid-prone landscapes retain moisture under increased drought severity. Many naturally-occurring minerals exhibit high surface areas compared with organic matter (Table 2), and this particular property has been explored to enhance soil moisture retention.

**Table 2.** Surface areas of a range of prospective materials for increasing soil moisture retention. Quartz (sand) is shown for a comparison against the high surface area materials.

| Material | Mineralogy | Specific Surface Area $m^2/g$ | Reference |
|---|---|---|---|
| Organic matter | NA | 60–480 | [92] |
| Zeolite | Mordenite | 1150 | [93] |
| Allophane | Pure | 1000 | [94] |
| Smectite | Montmorillonite | 750 | [95] |
| Vermiculite | Pure | 504 | [96] |
| Smectite | Montmorillonite | 230 | [97] |
| Zeolite | Clinoptilolite | 200 | [98] |
| Iron oxide | Goethite | 31 | [99] |
| Pumice | Sodium feldspar, pyroxene, olivine | 5–15 | [100] |
| Sand | Quartz, feldspar | 7.6 | [101] |

Mi et al. [102] showed that bentonite additions of up to 30 t/ha to a sandy loam increased soil water retention by almost 30% in an arid millet growing region of Northern China. Xiubin and Zhanbin [93] found zeolite (mordenite) application to a calcareous loess soil that increased moisture retention by up to 2% during drought and 15% under 'general' conditions, even though details on the zeolite application rate were not reported. Ippolito et al. [103] determined that adding clinoptilolite zeolite at a rate of up to 44 t/ha increased soil moisture retention by approximately 2% in a sandy textured loam supporting corn crops. A 1% to 2% percent increase may appear immaterial, but it effectively equates to several additional tonnes of soil water at the hectare scale. Kingston [13] likewise showed that adding varying amounts, up to 200 t/ha, of zeolite to common agricultural soils increased soil moisture retention from 1 to 2% under drought conditions in a glasshouse trial. Figure 3 shows that a zeolite application rate of 20 t/ha is sufficient to provide enough water (5 t/ha) to support a 50% increase in crop yield based on typical dry matter yields [104] and plant water contents [105]. Cost considerations are important at these relatively high applications rates, but these type of soil amendments may only need to be applied one-off or very infrequently compared with fertiliser applications.

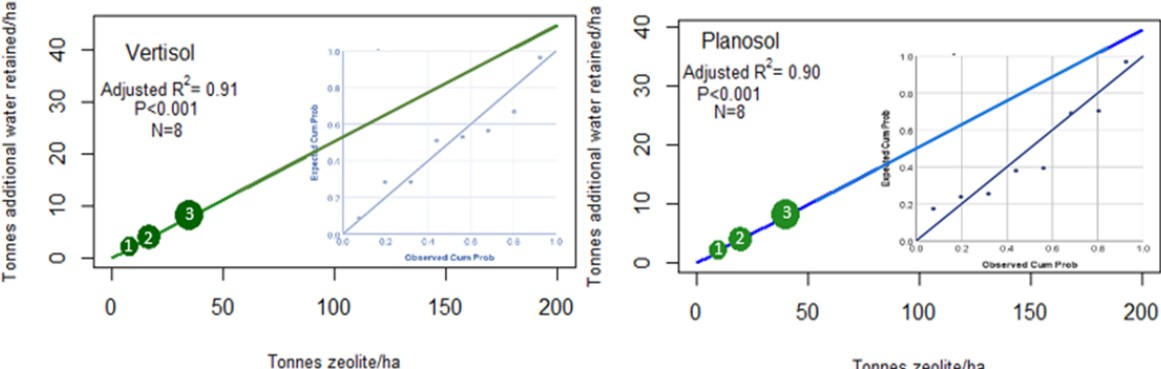

**Figure 3.** Example of how zeolite additions affect water retention for two soil types—Vertisol and Planosol—under drought conditions. Linear regression models applied to data from Kingston [13]. See methods for experimental details. Model fits plotted as insets for each soil type. Estimated tonnes of additional water provided by zeolite amendments derived using model outputs and assuming bulk density values of 1.6 $t/m^3$ for Vertisol [15] and 1.1 $t/m^3$ for Planosol [14], using a reference point of zero for soils with no zeolite amendments. Points '1', '2' and '3' represent soil moisture quantities needed to support 1.25×, 1.5× and 2× typical crop yields of 2 t/ha, reported by Anderson et al. [104], at average plant moisture content of 85% [105].

In addition to enhancing soil moisture, geo-amendments could be used to supply elements renowned for their capacity to strengthen plant structure and increase resilience to environmental stresses such as drought. A key element in this regard is silicon (Si). Silicon is now widely recognised for its diverse modes of supporting plant growth under drought conditions [106]. An increase in biogenic amorphous silica from 1 wt % to 5 wt % in soil can increase the plant-available water content by >60% [107]. Silicates are transported through roots with water as dissolved silicic acid and deposited in cell lumens, cell walls, and intercellular spaces of roots and leaves as amorphous hydrated silica ($SiO_2 \cdot H_2O$) [108]. He et al. [109] investigated the role of Si in the structural and chemical mechanisms of plants at a cellular level. They found Si was naturally present as a component of cell walls in suspended rice (*Oryza sativa*) cultures. Si was bound firmly to the cell wall matrix rather than as intra-cellular and extra-cellular deposits. The presence of Si in the cell wall matrix enhanced the maintenance of cell shape by improving the structural stability of the cell wall.

While further work at the molecular level is needed to fully understand Si-mediated alleviation of drought stress, physical, physiological, and biochemical mechanisms have been proposed, including reduction in oxidative stress, improved nutrient uptake, osmotic and phytohormonal adjustment, and modification of stomatal gas exchange [110]. Gong et al. [111] found that Si addition—in the form of sodium silicate—to wheat pots improved plant condition under drought and proposed an underlying mechanism related to an increase in antioxidant defence abilities. Silicon, added as potassium silicate, was shown to improve photosynthetic and transpiration rates in rice plants subjected to drought stress in a pot trial by Chen et al. [112]. Several driving processes were postulated including enhanced plant water status, an increase in the plant photosynthesis rate, and an improvement in mineral nutrient absorption [112].

Methods for increasing plant-available Si in soils generally involve application of harvested plant residues such as rice straw [113] or industrial by-products like steel slag [114]. Geomaterials could be pursued for their ability to deliver plant-available Si to soil where plant residues are not widely available in order to avoid potentially harmful environmental effects associated with industrial by-products. While most rocks contain >20% Si by weight, very few contain appreciable quantities of soluble Si. An exception is basalt, which is abundant and a potentially effective source of available Si. It contains high proportions of fast-weathering silicate minerals such as wollastonite, which was shown to be an effective agent for Si accumulation in rye grass in a study by Nanayakkara et al. [115]. Other geological candidates that could potentially deliver available Si include newly-formed clay and zeolite minerals, which both exhibit a high degree of isomorphous substitution whereby Si atoms are substituted from the aluminosilicate structure by aluminium (Al) [116]. These materials not only exhibit high surface areas for soil moisture retention, but also provide plant-available Si for further drought resilience.

Geo-amendments might also support crop viability under dry conditions through their light reflectance properties. The temperature of bare soils often exceeds 50 °C [117] and has been recorded to exceed 65 °C in arid landscapes [118]. Extreme temperatures pose a threat to plant health during the critical establishment phase through heat stress and water evaporation. Moreover, under warming climates, it is likely that the surface temperatures of barren agricultural soils will increase further. Techniques to alter soil temperature have involved plastic covers/mulches and aluminium surfaces even though these have typically been used to heat soil rather than cool it, which, often, depend on the colour and transparency of the plastic films in particular [119,120]. The use of these materials poses long-term threats to soil health such as microplastic contamination and increased water repellency [121].

Application of geo-materials might achieve the same outcome without adverse environmental impacts. Reflective minerals such as smectites, zeolites, and vermiculite could be surface-applied to barren cropping soils to help reduce soil temperature and retain soil moisture. Figure 4 depicts the results of an experiment exploring this application by comparing the surface temperature and light reflectance

properties of four common cropping soils with four reflective geo-materials exposed to ambient sunlight for a two-hour period.

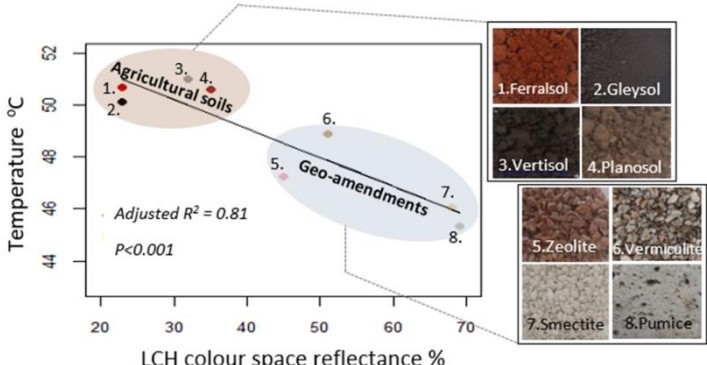

**Figure 4.** Relationship between surface temperature and LCH (lightness/reflectance, chroma, hue) colour space reflectance properties of four common cropping soils and four reflective geo-materials. The surface temperatures are averages taken over a 2-h period with an ambient air temperature of 37.1 °C.

Albeit preliminary, the results clearly show a significant trend between surface temperature and material reflectance as recorded in the LCH colour space. Hue and chroma—the other two dimensions of the LCH colour space model—did not relate well to surface temperature, which accords with the observations by Greer and Dole [119] who noted that brightness is likely more important than colour in regulating soil temperature. To our knowledge, modifying soil temperatures and associated moisture content with geo-amendments has not been investigated. Although the data in Figure 4 are promising, the and logistics of applying reflective materials to barren soils at paddock-scale would need to be tested to confirm the efficacy of this approach, including an evaluation of the confounding effects of other key factors on soil temperature (e.g., material composition, surface roughness, and moisture content).

## 6. Climate Change Mitigation

### 6.1. Carbon Dioxide

Basalts and ultramafic rocks are globally widespread and contain high amounts of iron and magnesium silicates—notably olivine and pyroxene—that are chemically unstable at the Earth's surface. The weathering of these minerals consumes atmospheric carbon dioxide and results in the precipitation of poorly-soluble carbonates and silicic acid. Equation (1) shows the weathering reaction of forsterite (the olivine Mg end-member) as an example.

$$Mg_2SiO_4 + 2CO_2 + 2H_2O \leftrightarrow 2MgCO_3 + H_4SiO_4 \tag{1}$$

The role of $CO_2$ in silicate weathering has long been recognized [122]. In the past two decades, researchers have identified the potential to harness the process to mitigate climate change [123]. Initially restricted to confined geo-engineering applications with value centred on industrially-useful by-products (e.g., iron oxides, carbonates), the approach has been recently appraised for agricultural contexts in the form of land-applying crushed volcanic rocks as a soil amendment.

Edwards et al. [11] reviewed the logistics of applying crushed basalt on tropical croplands as a global $CO_2$ mitigation strategy. The authors identify tropical regions as being ideally suited to the approach given the faster kinetics in chemical weathering resulting from the higher temperatures and rates of microbial activity in the tropics. Although the authors do not attempt to quantify the global $CO_2$ offsets

achieved by this approach, they note that it could be applicable to more than 680 million hectares of land. Given application rates would likely need to be 10 tonnes rock/ha to achieve soil conditioning benefits [11] and that most basalts contain approximately 20% by weight glass, olivine, and clinopyroxene (i.e., the readily weatherable mineral fraction of silicate rocks) [124], this would result in a $CO_2$ sequestration rate of 1 Gt/year across global agricultural lands based on the stoichiometry of Equation (1). This calculation assumes complete weathering of olivine-class minerals over a period of one year, complete efficiency in the weathering reaction, and the formation of irreversibly-insoluble carbonate minerals. It also does not account for the carbon footprint of crushed volcanic rock application to farmland, which Edwards et al. [11] estimate to be 20% of gross GHG reductions. With these considerations factored-in, it appears that enhanced weathering of crushed volcanic rocks in tropical farmland could mitigate approximately 1% of total global emissions, which are about 50 Gt $CO_2$ equivalents/year [125]. Although this magnitude may seem negligible, it is in the vicinity of total emission estimates for a number of industries considered to be 'big emitters' (e.g., cattle farming, rice production). Hence, Edwards et al. [11] cites the approach as a promising 'negative emission technology' while acknowledging potential downsides to the technique, which will be discussed later.

Kantola et al. [126] reported similar mitigation potential for enhanced basalt weathering in cropping areas of the USA, which range ranging from 0.2 to 1.1 Gt of $CO_2$/year. Similar to Edwards et al. [11], they also note the need to establish the risks associated with the technique. Strefler et al. [127] documented a $CO_2$ sequestration potential to 5 Gt/year for basalt applied to global croplands, even though this estimate is based on theoretical calculations and scale-up modelling.

Basalt application to acid soils presents an interesting scenario for effective $CO_2$ mitigation combined with pH neutralisation. Basalts can increase soil pH through hydroxyl ion production from silicate dissolution, development of secondary high cation exchange clay minerals from chemical weathering [128], removal of protons from solution by exchange during mineral weathering [129], and the development of carbonate mineral species. Moreover, under strongly acidic soil conditions, the kinetics of silicate mineral weathering could be significantly enhanced. Hausrath et al. [130] note that the solubility of olivine, which is the fastest-weathering silicate mineral and main source of secondary carbonate mineral precipitation, substantially increases with decreasing pH. Hangx and Spiers [129] report that the exchange of $H^+$ with $Mg^{2+}$ at the mineral surface is the controlling step in basalt weathering. These authors note that the rate of olivine dissolution is proportional to the square root of concentration of $H^+$ ions, which means that the olivine weathering rate effectively increases by 10-fold with a pH change from 6 to 4. This indicates that basalt weathering should be enhanced, at least initially, in acid-affected soils. For future research, when testing basalt dissolution rates in acid soils, the subsequent effect on pH and associated formation of carbonate mineral phases is recommended, as verified by chemical and X-ray diffraction (XRD) techniques.

It is important to note that the practicality of the types of net emission reduction approaches discussed in this paper will be highly dependent on detailed cost benefit analyses. These approaches will likely be sensitive to drivers such as associated benefits (e.g., additional nutrient supply, intrinsic drought resilience) and government/industry incentives for GHG mitigation.

### 6.2. Methane

Methane ($CH_4$) is the second greatest contributor to climate change at an emission rate of approximately 8 Gt $CO_2$-equivalents/year with agricultural sources responsible for almost half of these emissions [131]. One of the most significant agricultural $CH_4$ emitting activities is rice farming, which produces about 1.25 Gt $CO_2$-e/year [125]. Iron oxides, which are a ubiquitous and diverse class of geological minerals, have shown promise to mitigate $CH_4$ emissions produced by this industry. Since these minerals are powerful electron acceptors, their presence in anaerobic environments allows iron-reducing microorganisms

to outcompete methanogens, which utilise the less energetically-favourable pathway of $CO_2$ electron acceptors. Watanabe and Kimura [132] demonstrated the importance of soil ferric iron content in regulating $CH_4$ emissions in paddy soils. Numerous researchers have subsequently sought to exploit this process by applying granular iron oxides to inundated rice paddies.

In a glasshouse study, Jäckel and Schnell [133] showed that application of the iron oxide ferrihydrite to soil at a rate of 30 g/kg decreased $CH_4$ emissions by 84% without any impact on rice crop yield. It is worth noting that this ferrihydrite application was quite high at approximately 45 t/ha (derived from 'Method' details). Ali et al. [134] tested lower iron oxide application rates—up to 4 t/ha—to rice paddy soils in South Korea. Using steel slag as the iron oxide source, these authors reported a 20% decrease in $CH_4$ emissions over a year-long field trial [134]. They also documented a 13–18% increase in rice crop yield in slag-amended plots and noted that iron oxides possess soil conditioning benefits in addition to $CH_4$ mitigation potential [134]. In a field trial in China, Wang et al. [135] demonstrated that iron oxide addition (as steel slag) to rice paddies resulted in a 49% decrease in $CH_4$ emission at an application rate of 8 t/ha.

Sulphates—including gypsum ($CaSO_4 \cdot 2H_2O$), thenardite ($Na_2SO_4$), and mascagnite (($NH_4)_2SO_4$)—are another class of commonly-occurring minerals that have been investigated by mitigating rice paddy $CH_4$ emissions [136]. Sulphate applications not only have the potential to suppress $CH_4$ emissions, they also perform the role of soil sulphur fertilisation. Lindau et al. [137] reported that gypsum addition to rice fields in Louisiana at application rates of 2 t/ha suppressed $CH_4$ emissions by 46% over a 70-day period. Minamikawa et al. [138], using a regression design, showed that application rates of $(NH_4)_2SO_4$ up to 135 kg/ha decreased $CH_4$ emissions by approximately 85% in rice fields in Japan.

A variety of electron acceptor minerals clearly have potential to mitigate $CH_4$ emissions from rice farming. Key unresolved issues surrounding the approach include ascertaining the impacts of the amendments on crop yield and a cost benefit analysis of the technique, which will require resolving the period of effectiveness of the treatments. In the case of sulphate additions, this likely continues until the by-product nutrients are biologically consumed. For iron oxides, the duration of effectiveness may continue over multiple seasons depending on the kinetics of iron reduction.

*6.3. Nitrous Oxide*

Nitrous oxide is the third-ranked contributor to climate change, representing 5% of annual anthropogenic emissions [131]. Land-applied nitrogen fertilisers are the main $N_2O$ emission source. Harnessing geological materials to mitigate $N_2O$ emissions is a reasonably new concept, and began with the recognition that charged minerals (e.g., clays and zeolites) can suppress ammonium ($NH_4^+$) losses from fertilised soils and act as slow-release deliverers of N to crops [139–141]. In fact, protection of $NH_4^+$ loss is itself a mechanism to mitigate $N_2O$ emissions because ammonium is an indirect GHG with an estimated 1% of re-deposited $NH_4^+$ converting to $N_2O$ [131].

Zaman and Nguyen [142] identified the potential for zeolites to mitigate $N_2O$ emissions from fertilised pastures in New Zealand. In a 56-day field trial, these authors found that urea application to soil in the presence of zeolite (at an application rate of 35 t/ha—calculated from the author's description of 1 kg zeolite needed to retain 5.7 g of $NH_4^+$-N) resulted in an 11% decrease in $N_2O$ emissions. The $N_2O$ emission decrease was attributed to $NH_4^+$ exchange on the negatively-charged zeolite and subsequent nitrification suppression [142].

Hill et al. [143] investigated the potential for vermiculite—a high cation exchange capacity clay—to decrease $N_2O$ emissions from organic and chemical N fertilisers applied to a range of soils in a glasshouse trial. These authors reported an average 70% decrease in $N_2O$ emissions at a vermiculite application rate of 7 t/ha over a one-year period. Venterea et al. [22] investigated the specific mechanisms by which high cation exchange capacity (CEC) materials suppress $N_2O$ emissions by assessing soil's physical, chemical,

and genomic properties and emission rates under fertiliser application. They concluded that high CEC materials retain $NH_4^+$, which prevents accumulation of $NH_3$ to toxic levels within nitrite-oxidising bacterial communities, and, thereby, enables nitrification to proceed freely without nitrite build-up producing high $N_2O$ emissions.

*6.4. Integrating Geo-Based Climate Change Mitigation with Established Practices*

The geo-based climate change mitigation technologies discussed above could be integrated with agricultural GHG abatement practices whose effectiveness have been previously demonstrated. This offers a significant opportunity to realise broad-scale benefits with the Food and Agriculture Organisation of the United Nations/FAO [144] noting that "agriculture is the only sector that has the capacity to remove GHGs safely and cost-effectively from the atmosphere without reducing productivity." Proven agricultural GHG emission reduction practices centre around techniques to bolster soil carbon stocks and include low/zero tillage, erosion minimisation, conservation agriculture, and returning crop residues to agricultural land [145]. Smith and Olesen [146] estimate that these farming practices could achieve GHG emission reduction of up to 4.2 $CO_2$-e Gt/year. By comparison, geo-based technologies could mitigate approximately 2 Gt $CO_2$-e annually, based on average performance efficiencies reported in the previous sections. Established practices and geo-technologies combined might, therefore, offer a sizable 12% mitigation potential for total anthropogenic emissions.

## 7. Risks and Environmental Concerns

Geological inputs into soils to rejuvenate agricultural production systems entails several potential risks. Many of these are well-summarised by Edwards et al. [11] in their review of enhanced rock weathering to mitigate $CO_2$ emissions and add nutrient value to tropical farmland. Broadly, they identify five major risks associated with enhanced weathering approaches: 1. an increased GHG footprint associated with extracting, transporting, and processing materials, 2. leaching of toxic trace metals, 3. biodiversity impacts adjoining farms, 4. impacts to water quality in receiving catchments, and 5. an increased mining footprint [11].

Land degradation and GHG footprints associated with increased mineral extraction activities can be estimated based on the scale of proposed operations. Leaching of toxic metals is only really a concern for application of ultramafic peridotite minerals with Ni and Cr being the main candidate risks [147]. The most common silicate rocks, which form a crystallisation continuum between basalt and felsic granites, generally do not contain any trace metals above typical background soil concentrations [17]. Contamination of waterways by washed down rocks and minerals as well as subsequent impacts to water quality and biodiversity are serious risks associated with increased geological inputs to agricultural soils. Yet, it is difficult to predict the consequences. Areas at greater risk will likely involve farmlands receiving high rainfall in rugged terrain. However, increased mineral inputs into soils may not necessarily be linearly related to increased sediment outputs during rainfall. For example, Thornton et al. [148] note that farm management practices, such as cover cropping, exert a considerable influence on erosion risk. Ultimately, determining regions at risk of adverse water quality impacts could be evaluated using global cropland erosivity databases, such as those produced by Panagos et al. [149] in combination with on-ground farm management practice information.

Other potential drawbacks to increased geological inputs into agricultural landscapes at a broader scale include triggering greater net GHG emissions (for example, $N_2O$ production overshadowing $CH_4$ mitigation in rice fields from iron oxide amendments, as noted by Huang et al. [150]), and an increased risk of silicosis and other respiratory diseases associated with quarrying and processing silicate rocks and minerals [151]. A further, overlooked potential risk associated with increased geological amendments to

soils is large scale changes to the geo-microbial community structures of agricultural landscapes. Native soil microbial dynamics have been altered via agricultural practices through physical disturbances to the landscape as well as ongoing chemical inputs [152].

Sustained changes to the inorganic mineral structure of agricultural soils might further alter microbial dynamics. This is especially likely with ongoing additions of materials that have high surface areas, such as zeolites and clays, because mineral surfaces are important sites for soil microbial communities [153,154]. Considering that a typical agricultural Planosol has a surface area of approximately 10 $m^2$/g [155], additions of zeolites with a surface area of 1000 $m^2$/g [93] at a rate of 20 t/h/year would result in the soil surface area increasing to 100 $m^2$ after five years of sustained zeolite addition (assuming a soil bulk density of 1.1 $t/m^3$ [14] and an active incorporation depth extending to the base of the topsoil, i.e., 0.2 m). Clearly, the potential for soil geological amendments to alter soil microbial communities in agricultural landscapes is high, and potential downsides to this need to be considered in conjunction with any realised benefits.

## 8. Outlook

The converging threats of climate change and soil degradation will necessitate large-scale shifts to current agricultural practices. Geological amendments can help streamline this transition. Certainly, in terms of availability, many of the geo-materials discussed in this paper are widely abundant (Figure 5), which bodes well for practical uptake in the agricultural sector. Moreover, most of the rocks and minerals that can support agriculture are reasonably low cost (approximately AUD (Australian dollars) $200/t—excluding transport) when compared with most fertiliser inputs (urea and ammonium phosphate are about AUD $400/t—excluding transport [156]). Additionally, unlike fertiliser input requirements that are ongoing, some landscapes may only require one-off or sporadic inputs of geo-amendments.

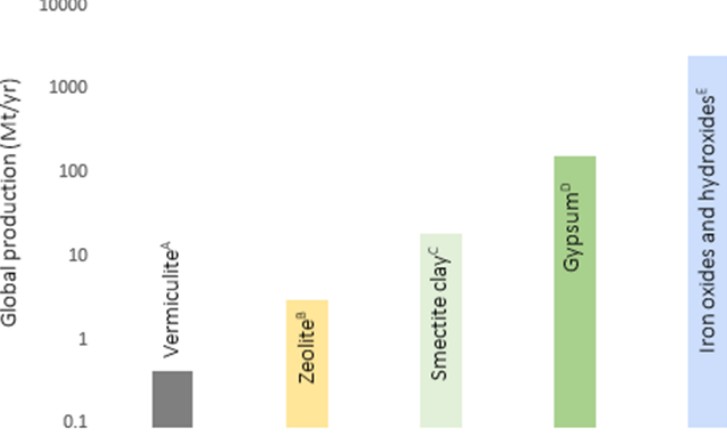

**Figure 5.** Global production quantities for main minerals reviewed in this paper. Note: these are estimated production quantities not reserves, as USGS (United States Geological Survey) reports that reserve estimates for most minerals are not known. References: A = [157], B = [158], C = [159], D = [160], and E = [161].

In Figure 6, we highlight how the key mineral groups discussed in this manuscript can assist in addressing the key agricultural challenges. Various options exist regarding how a concerted 'geo-agriculture' approach could be developed. Addition of minimally-processed geo materials to cropland is the simplest and cheapest option. Some minerals can support a requirement to address single threats to agriculture, such as olivine, which can sequester $CO_2$ emissions and mitigate against drier and warmer growing conditions (Figure 6). Others have the potential to combat multiple threats, such as zeolites and iron oxides,

which can reduce nutrient contamination of waterways in agricultural catchments as well as improve soil climate resilience (Figure 6).

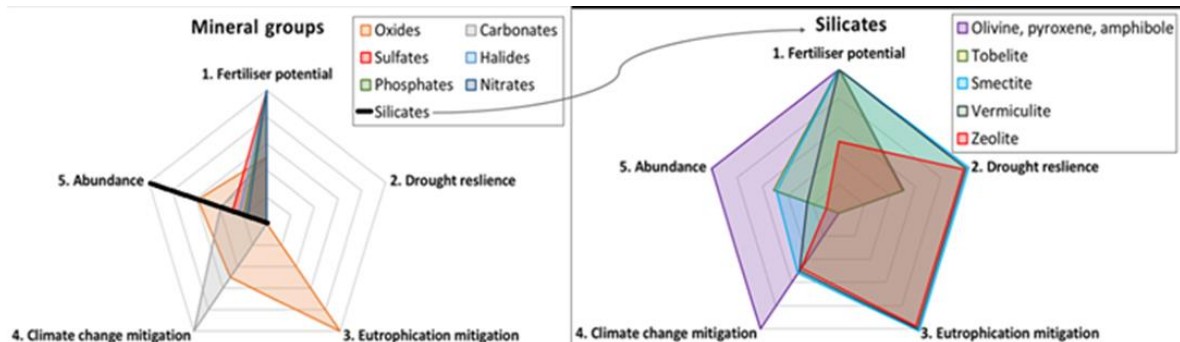

**Figure 6.** Semi-quantitative expression of key agricultural attributes for the main mineral groups. Note: abundance rankings derived from elemental and mineral crustal abundance data in Reference [17] and Hartmann and Moosdorf [162].

Looking further ahead, the development of engineered formulations of geological materials with other compounds may offer wider-reaching benefits. This approach will involve greater input costs, but could ultimately prove to be the most effective way to rejuvenate agricultural production systems in the future. It may be possible to develop bespoke geo-based formulations that can tackle nearly all of the big challenges for attaining viable agricultural production. We propose two potential forms that a 'complete-performing' formulation might take. The first is a nutrient-doped zeolite/clay or hydroxide granule, using waste streams [57] or agricultural runoff [81] as the nutrient 'seed'. The high-surface area and chemically-charged mineral component of the granules will simultaneously offer drought resilience, eutrophication protection, and climate change mitigation via $NH_4^+$ retention and decreased $N_2O$ emissions [22]. Moreover, the nutrients adsorbed to the granules could offer sustainable fertiliser supply through ion exchange processes in the rhizosphere. The mineral granules in this proposed formulation will need to be ground or milled (comminution) to a size that is effective for both the nutrient-doping and application stage with consideration toward promoting granule retention in the soil. Since comminution is a very expensive and inefficient operation [163], minimising processing requirements for these formulations will be a crucial step in developing an economically viable product.

A second potential 'complete-performing' formulation could take the form of a blend of microbial inoculum and rapid-weathering silicate minerals (i.e., from the olivine-pyroxene-amphibole series). In addition to the rock phosphate solubilising bacteria discussed previously, a range of bacterial consortia have been demonstrated to actively breakdown silicate rocks [164]. These microbial assemblages could be integrated with silicate granules and introduced into agricultural soils. Successful incorporation of bacterial assemblages—for functions such as pathogen resistance and phosphate solubilisation—has been recently undertaken in agricultural landscapes [165,166]. In these cases, the bacterial populations are typically grown on an organic, agar medium and then dried and pelletised. For bacterial consortia to be blended with geological materials, a linking medium will likely be required. Biodegradable and cheap polymer materials, such as polysaccharide gums, which have been specifically tested as soil amendments [167,168], could act as ideal carriers for linking bacterial isolates with fast-weathering silicate granules. This type of formulation would offer benefits across each of the four key themes discussed in this review. The bacteria-accelerated silicate breakdown will provide plant available nutrients and secondary clays with high water and nutrient retention capacities as well as generate stable carbonate mineral phases to assist in mitigating agricultural GHG emissions. Testing the performance of this type of formulation is needed along with appraisal

of associated engineering costs including polymer extrusion/formulation and comminution of mineral granules. Nonetheless, any development successes in this area will undoubtedly reduce overall reliance on conventional fertiliser products, and, thereby, rejuvenate viable agricultural practices as the sector faces new challenges to be independently sustainable under emerging pressures from climate change.

## 9. Conclusions

Geo-based technologies can contribute to several emerging challenges in the agricultural sector. These include offering new viable fertiliser sources, especially for N and P, preventing environmental contamination in the form of eutrophication, which is combined with the prospect of returning up to 10% of N and P lost from agricultural land, assisting crops to cope with drought stress, and mitigating climate change by potentially offsetting approximately 5% of the total annual emissions, and up to 12% when integrated with proven farm management practices such as reduced tillage and crop rotation. Incorporating geo-amendments into agricultural practices could span a spectrum of investment levels, which range from land application of minimally processed (i.e., crushed) rocks and minerals to bespoke engineered products such as nutrient-doped smectites/zeolites/oxides and polymer encapsulated microbial inoculum/mineral formulation blends. Ultimately, a 'geo-agriculture' approach has the potential not only to assist in meeting some of the key challenges facing the agricultural sector, but also to bring about wider societal benefits that align with the United Nations Sustainable Development Goals.

**Author Contributions:** C.P.—instigator and primary author. K.K.—contributions on drought resilience. B.L.—contributions on plant-microbial interactions. I.L.—contributions on plant-Si interactions and Figure 6 creator. S.P.—contributions on wastewater P recovery. All authors have read and agreed to the published version of the manuscript.

**Funding:** This research received no external funding.

**Acknowledgments:** Thanks to David Hamilton of the Australian Rivers Institute for proof-reading and suggestions. Thanks to the CRDC for student support towards this project.

**Conflicts of Interest:** The authors declare no conflict of interest.

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
