# Peer review of "Geo-Agriculture: Reviewing Opportunities through Which the Geosphere Can Help Address Emerging Crop Production Challenges"

_agronomy, doi:10.3390/agronomy10070971_

Round 1
Reviewer 1 Report
The theme approached is important and interesting in the daily life because of climate change. Reducing contamination, increasing the drought resilience, mitigating GHGs emissions from the atmosphere, in the end, all that means improving soil health by considering the geosphere, and acknowledging how it operates.
Keywords are missing
The Introduction sets the scene quite well in the sense of the interesting objectives to be achieved.
Methods
Line 82. Plots?
Line 108. There is a final point missing.
Line 110. I would suggest authors to increase the figure 1 size in order to be more legible.
Line 110. The figure should have a higher quality.
Line 110. I cannot read the “y” axe of the last part of the figure (micronutrient elements).
Point 4 and 5.
Line 305. The figure 2 is also repeated at the end of page 8 (point 5).
Line 334. Table 2. There is a word highlighted.
Line 345. There is a final point missing.
Line 372. Should not be plots? Instead of pots?
Line 423. Point Carbon dioxide. Best management practices in agriculture such as zero tillage, minimum tillage, should be also considered within this part as a valuable tool to mitigate climate change in permanent and in annual crops. Examples such as Conservation agriculture. There are many references in FAO, etc.
Although the paper is kind of review, there should be in my view a conclusion, where to highlight the most valuables ways in which geosphere can make the difference for agricultural challenges (Climate change, etc.)
The theme is a challenge itself not only for agriculture but also for society in general.
There many references, and some of them are very recent.
The numbers are repeated.
Reference 133 is missing
Author Response
|
Reviewer #1 |
|
|
|
Comment |
Authors’ response |
New text (in red) in revised MS
|
|
The theme approached is important and interesting in the daily life because of climate change. Reducing contamination, increasing the drought resilience, mitigating GHGs emissions from the atmosphere, in the end, all that means improving soil health by considering the geosphere, and acknowledging how it operates. Keywords are missing
|
We thank the reviewer for their appraisal of our manuscript and their constructive suggestions.
Keywords have been placed into the revised submission. |
|
|
Line 82. Plots?
|
In this case they were pots (1L) used in a glasshouse trial. We have clarified in the revised MS. |
|
|
Line 108. There is a final point missing. |
Corrected in revised MS. |
|
|
Line 110. I would suggest authors to increase the figure 1 size in order to be more legible. |
We have modified Fig. 1 according to the Reviewer’s suggestion. The axis units have been removed from the panels and instead referred to in the caption. The size of the axis titles has been increased. |
|
|
Line 110. The figure should have a higher quality. |
Fig 1 has been converted to a much higher resolution file type (1800 kb compared with 300 kb previously), as per the reviewer’s suggestion. |
|
|
Line 110. I cannot read the “y” axe of the last part of the figure (micronutrient elements). |
In response to the Reviewer’s comment, the axes units have been removed from the panels and instead referred to in the caption in the revised MS. |
Lines 171-178… “Figure 1. – Typical concentrations of essential plant i) macro (values in wt%) and ii) micro elements (values in ppm, except iron which is wt%) across a range of prospective geological sources (“minerals” and “rocks”) for use in agriculture, with conventional fertiliser contents (“fertilisers”) for macro-elements shown for comparison. Fertiliser and mineral concentrations derived from chemical stoichiometry. DAP=diammonium phosphate, MAP=monoammonium phosphate, Amm. Smect.= ammonium smectite, Triple Super P = Triple Super Phosphate, Ig. Phosph.=Igneous Phosphorites, Sed. Phosph=Sedimentary Phosphorites. Reference sources: A=[16], B=[17], C=[18], D=[19], E=[20], F=[21].”
|
|
Line 305. The figure 2 is also repeated at the end of page 8 (point 5). |
The second Fig 2 has been removed from the revised MS. |
|
|
Line 334. Table 2. There is a word highlighted. |
In response to the Reviewer’s comment, the highlighting has been removed from this table. |
|
|
Line 345. There is a final point missing. |
Corrected in revised MS. |
|
|
Line 372. Should not be plots? Instead of pots? |
In this case, the study was conducted using pots rather than plots. |
|
|
Line 423. Point Carbon dioxide. Best management practices in agriculture such as zero tillage, minimum tillage, should be also considered within this part as a valuable tool to mitigate climate change in permanent and in annual crops. Examples such as Conservation agriculture. There are many references in FAO, etc. |
This is a good point and we have now included a new sub-section at the end of the Climate Change Mitigation section highlighting how geo-amendments could be integrated with proven agricultural mitigation practices to achieve substantive emission reductions. |
Lines 588-600….”6.4. Integrating geo-based climate change mitigation with established practices The geo-based climate change mitigation technologies discussed above could be integrated with agricultural GHG abatement practices whose effectiveness have been previously demonstrated. This offers a significant opportunity to realise broad-scale benefits, with the FAO [144] noting that “agriculture is the only sector that has the capacity to remove GHGs safely and cost-effectively from the atmosphere without reducing productivity”. Proven agricultural GHG emission reduction practices centre around techniques to bolster soil carbon stocks and include low/zero tillage, erosion minimisation, conservation agriculture and returning crop residues to agricultural land [145]. Smith and Olesen [146] estimate that these farming practices could achieve GHG emission reduction of up to 4.2 CO2-e Gt/yr. By comparison, geo-based technologies could mitigate approximately 2 Gt CO2-e annually, based on average performance efficiencies reported in the previous sections. Established practices and geo-technologies combined might therefore offer up to a sizable 12% mitigation potential for total anthropogenic emissions.” |
|
Although the paper is kind of review, there should be in my view a conclusion, where to highlight the most valuables ways in which geosphere can make the difference for agricultural challenges (Climate change, etc.). |
We have taken the Reviewer’s suggestion on-board and added a Conclusions section to the paper highlighting the key benefits that geological amendments can offer the ag sector in the context of emerging challenges. |
Lines 710-723…. “9. Conclusions Geo-based technologies can contribute to several emerging challenges in the agricultural sector. These include: offering new viable fertiliser sources, especially for N and P; preventing environmental contamination in the form of eutrophication, combined with the prospect of returning up to 10% of N and P lost from agricultural land; assisting crops to cope with drought stress; and mitigating climate change – potentially offsetting approximately 5% of total annual emissions and up to 12% when integrated with proven farm management practices, such as reduced tillage and crop rotation. Incorporating geo-amendments into agricultural practices could span a spectrum of investment levels: ranging from land application of minimally processed (i.e., crushed) rocks and minerals, through to bespoke engineered products such as nutrient-doped smectites/zeolites/oxides and polymer encapsulated microbial inoculum/mineral formulation blends. Ultimately, a ‘geo-agriculture’ approach has the potential not only to assist in meeting some of the key challenges facing the agricultural sector, but also to bring about wider societal benefits that align with the United Nations Sustainable Development Goals.” |
|
The theme is a challenge itself not only for agriculture but also for society in general. |
This is a good point, and incidentally was also identified by the other reviewer. In response, we have made additions to the text highlighting that geo-inputs into agricultural systems can bring about wider societal benefits beyond the agricultural sector. |
Lines 35-38…. “Our review highlights potential for ‘geo-agriculture’ approaches to not only help meet several key emerging challenges that threaten sustainable food and fiber production, but also to contribute to achieving some of the United Nations Sustainable Development Goals – ‘Zero Hunger’, ‘Life on Land’ and ‘Climate Action’.” Lines 77-82…. “Importantly, a number of the themes that we address in this paper highlight opportunities to realise benefits beyond the agricultural sector alone. Indeed, efforts to minimize pollution, foster resilient landscapes and combat emissions contributing to climate change are all wide-reaching societal aspirations. We identify that ‘geo-agricultural’ approaches could directly help support three – of the seventeen Sustainable Development Goals prioritized by the United Nations, viz. ‘Zero Hunger’, ‘Life on Land’ and ‘Climate Action’..” Lines 720-723…. “Ultimately, a ‘geo-agriculture’ approach has the potential not only to assist in meeting some of the key challenges facing the agricultural sector, but also to bring about wider societal benefits that align with the United Nations Sustainable Development Goals.”
|
|
The numbers are repeated. Reference 133 is missing |
We have corrected the reference numbering in the revised MS. |
|
Reviewer 2 Report
In the times of changing paradigm of agriculture from industrial oriented towards more non-industrial sustainable and agroecological approaches the reviewed paper is an important contribution. It highlights the opportunities that provide naturally occuring rocks and minerals not only for agriculture production, but describe also a wider perspective in relations to contemporary occuring problems i.e. of climate change.
The goal is well designed, however its justification is not suficient and need additional explanation, especially with relation to challanges that face not only agricultral sector, i.e. Sustainable Development Goals.
Altough the Method section is suficiently explained there are missing two key elements: 1. How the literature review was executed, and 2. How the semi-quantitative analysis presented in Figure 6 (l. 595) was performed.
The section 3-6 are quite clearly explained.
The section 7 – Risks and environmental concerns is underdeveloped. It contains very selective literature evidences about possible negative impacts. These need to be more in details explained and justified.
Also the section 8 – Outlook is very reductive and the discussion of prospects for futher development is presented by Authors only in 10 lines (603-613) and need further explanations adn justifications.
Overally I like the paper very much, therefore suggest minor, not major review, however, hope that they will take into account my suggestions for developing the paper by showing in details not only opportunities but also explaining in details threats along with perspective not only why geosphere can help address emerging crop production challenges, but also how through it can contribute to solving broader problems.
Author Response
|
Reviewer #2 |
|
|
|
Comment |
Authors’ response |
New text (in red) in revised MS
|
|
In the times of changing paradigm of agriculture from industrial oriented towards more non-industrial sustainable and agroecological approaches the reviewed paper is an important contribution. It highlights the opportunities that provide naturally occuring rocks and minerals not only for agriculture production, but describe also a wider perspective in relations to contemporary occuring problems i.e. of climate change. The goal is well designed, however its justification is not suficient and need additional explanation, especially with relation to challanges that face not only agricultral sector, i.e. Sustainable Development Goals.
|
We thank the reviewer for their appraisal of our manuscript and suggestions for improvement.
We take on-board the point that the justification needs additional explanation, in relation to global sustainable challenges – in fact, the other reviewer also picked up on this. In response, we have added new text in the Abstract, Introduction and the new Conclusions section to contextualise our work relative to global sustainability goals.
|
Lines 35-38…. “Our review highlights potential for ‘geo-agriculture’ approaches to not only help meet several key emerging challenges that threaten sustainable food and fiber production, but also to contribute to achieving some of the United Nations Sustainable Development Goals – ‘Zero Hunger’, ‘Life on Land’ and ‘Climate Action’.” Lines 77-82…. “Importantly, a number of the themes that we address in this paper highlight opportunities to realise benefits beyond the agricultural sector alone. Indeed, efforts to minimize pollution, foster resilient landscapes and combat emissions contributing to climate change are all wide-reaching societal aspirations. We identify that ‘geo-agricultural’ approaches could directly help support three – of the seventeen Sustainable Development Goals prioritized by the United Nations, viz. ‘Zero Hunger’, ‘Life on Land’ and ‘Climate Action’..” Lines 720-723…. “Ultimately, a ‘geo-agriculture’ approach has the potential not only to assist in meeting some of the key challenges facing the agricultural sector, but also to bring about wider societal benefits that align with the United Nations Sustainable Development Goals.”
|
|
Altough the Method section is suficiently explained there are missing two key elements: 1. How the literature review was executed, and 2. How the semi-quantitative analysis presented in Figure 6 (l. 595) was performed. |
We have added two new sub-sections in the Methods in response to this comment. |
Lines 85-99…. “2.1. Literature review The review was conducted following a systematic approach using Google Scholar and Web of Science databases. Initial key word searches included combinations of the terms: ‘agriculture’; ‘climate change’; ‘contamination’; ‘drought’; ‘fertiliser’; ‘geology’, ‘minerals’ and ‘rocks’. Based on the authors’ research and professional experience, four key ‘geo-agriculture’ opportunity themes were developed and searched: 1. New fertiliser prospects (key search terms included combinations of: ‘crustal abundance elements’, ‘high nitrogen and phosphorus content rocks and minerals’ and ‘waste nutrient streams’); 2. Environmental contamination prevention (key search terms included combinations of: ‘zeolites’, ‘clays’, ‘bentonite’, ‘oxides/oxyhydroxides’, ‘allophane’, ‘hydrotalcite’, ‘ammonia/ammonium’, ‘nitrate’, ‘nitrite’, ‘phosphate’ and ‘eutrophication’); 3. Drought resilience (key search terms included combinations of: ‘water holding capacity’, ‘zeolites’, ‘clays’, ‘bentonite’, ‘specific surface area’, ‘silicon’, ‘potassium’, ‘light reflectance’, and ‘temperature’), and 4. Climate change mitigation (key search terms included combinations of: ‘enhanced weathering’, ‘silicates’, ‘olivine’, ‘iron oxides’, ‘zeolites’, ‘clays’, ‘carbon dioxide’, ‘methane’, ‘nitrous oxide’, ‘ammonia’, and ‘agricultural climate change mitigation’).” Lines 127-143…. “2.4. Semi-quantitative analysis of benefical mineral attributes Towards the end of this review, in Figure 6, we present a semi-quantitative analysis of the four key mineral attributes reviewed. Indices were assigned according to the following criteria: 1. Fertiliser potential – the maximum value (outer contour band of web plot) was attributed if the mineral contributes at least one of the six macronutrient elements, median value (middle contour band of web plot) if mineral contributes micronutrient element or pH adjustment; 2. Drought resilience – the maximum value was assigned for minerals that increase moisture-holding capacity, median value scored if the mineral offers available forms of drought-protective elements (Si and K) or temperature protection via heat reflectance; 3. Eutrophication mitigation – given the maximum value if the mineral can sequester N and P compounds, otherwise naught was assigned; 4. Climate change mitigation – the maximum score was assigned for minerals with the potential to mitigate CO2 and median value assigned for minerals with potential to mitigate lower-magnitude GHGs (CH4 and N2O); and 5. Abundance – the following ranking values were assigned for the main mineral groups: 1. silicates, 2. oxides, 3. carbonates, 4. sulfates, 5. halides, 6. phosphates and 7. nitrates. The silicate subset groups were ranked as: 1. neso and ino-silicates, 2. smectite and vermiculite, 3. tobelite, and 4. zeolite. These rankings were then normalized to give the five (i.e., 20%) contour bands in the web plots..”
|
|
The section 7 – Risks and environmental concerns is underdeveloped. It contains very selective literature evidences about possible negative impacts. These need to be more in details explained and justified. |
We concede that this section was originally very brief, with reference to a single previous review. In response to the Reviewer’s comment we have expanded this section with additional references to expand on the potential downsides of the approaches discussed in the paper. |
Lines 610-624…. “Land degradation and GHG footprints associated with increased mineral extraction activities can be estimated based on the scale of proposed operations. Leaching of toxic metals is only really a concern for application of ultramafic peridotite minerals, with Ni and Cr the main candidate risks [147]. Indeed, the most common silicate rocks, which form a crystallisation continuum between basalt and felsic granites, generally do not contain any trace metals above typical background soil concentrations [17]. Contamination of waterways by washed down rocks and minerals, and subsequent impacts to water quality and biodiversity, are serious risks associated with increased geological inputs to agricultural soils – yet, it is difficult to predict the consequences. Areas at greater risk will likely involve farmlands receiving high rainfall in rugged terrain. However, increased mineral inputs into soils may not necessarily be linearly related to increased sediment outputs during rainfall. For example, Thornton et al. [148] note that farm management practices, such as cover cropping, exert a considerable influence on erosion risk. Ultimately, determining regions at risk of adverse water quality impacts could be evaluated using global cropland erosivity databases such as those produced by Panagos et al. [149] in combination with on-ground farm management practice information.”
|
|
Also the section 8 – Outlook is very reductive and the discussion of prospects for futher development is presented by Authors only in 10 lines (603-613) and need further explanations adn justifications. |
As with the point above, we concede that this section was quite limited in the original submission. In response, we have expanded this section by proposing two forms on potential formulations that could be developed. We have also added additional reference material to this section to highlight probable logistical constraints that are associated with our proposed formulations. |
Lines 673-709…. “Looking further ahead, the development of engineered formulations of geological materials with other compounds may offer wider-reaching benefits. This approach will involve greater input costs but could ultimately prove to be the most effective way to rejuvenate agricultural production systems in the future. Indeed, it may be possible to develop bespoke geo-based formulations that can tackle nearly all of the big challenges for attaining viable agricultural production. We propose two potential forms that a ‘complete-performing’ formulation might take. The first is a nutrient-doped zeolite/clay or hydroxide granule, using waste streams [57] or agricultural runoff [81] as the nutrient ‘seed’. The high-surface area and chemically-charged mineral component of the granules will simultaneously offer drought resilience, eutrophication protection, and climate change mitigation – via NH4+ retention and decreased N2O emissions [22]. Moreover, the nutrients adsorbed to the granules could offer sustainable fertiliser supply through ion exchange processes in the rhizosphere. The mineral granules in this proposed formulation will need to be ground or milled (comminution) to a size that is effective for both the nutrient-doping and application stage, with consideration to promoting granule retention in the soil. As comminution is a very expensive and inefficient operation [163] minimising processing requirements for these formulations will be a crucial step in developing an economically viable product. A second potential ‘complete-performing’ formulation could take the form of a blend of microbial inoculum and rapid-weathering silicate minerals (i.e., from the olivine-pyroxene-amphibole series). In addition to the rock phosphate solubilising bacteria discussed previously, a range of bacterial consortia have been demonstrated to actively breakdown silicate rocks [164]. These microbial assemblages could be integrated with silicate granules and introduced into agricultural soils. Successful incorporation of bacterial assemblages – for functions such as pathogen resistance and phosphate solubilisation – has been recently undertaken in agricultural landscapes [165, 166]. In these cases, the bacterial populations are typically grown on an organic, agar medium and then dried and pelletised. For bacterial consortia to be blended with geological materials, a linking medium will likely be required. Biodegradable and cheap polymer materials, such as polysaccharide gums, which have been specifically tested as soil amendments [167, 168], could act as ideal carriers for linking bacterial isolates with fast-weathering silicate granules. This type of formulation would offer benefits across each of the four key themes discussed in this review – bacteria-accelerated silicate breakdown will provide plant available nutrients and secondary clays with high water and nutrient retention capacities, as well as generate stable carbonate mineral phases to assist in mitigating agricultural GHG emissions. Testing of the performance of this type of formulation is needed along with appraisal of associated engineering costs, including polymer extrusion/formulation and comminution of mineral granules. Nonetheless, any development successes in this area will undoubtedly reduce overall reliance on conventional fertiliser products, thereby rejuvenating viable agricultural practices as the sector faces new challenges to be independently sustainable under emerging pressures from climate change..”
|
|
Overally I like the paper very much, therefore suggest minor, not major review, however, hope that they will take into account my suggestions for developing the paper by showing in details not only opportunities but also explaining in details threats along with perspective not only why geosphere can help address emerging crop production challenges, but also how through it can contribute to solving broader problems. |
We thank the reviewer for their overall positive appraisal of our work. We have taken on-board their constructive suggestions and hope they find it has strengthened and broadened the appeal of our paper. |
|
Round 2
Reviewer 1 Report
After checking all my fisrt suggestions and the second version of the manuscript provided by the authors, I think this research work ahs been ammended.